# Single pixel imaging at megahertz switching rates via cyclic Hadamard masks

Evgeny Hahamovich[1,2], Sagi Monin [1,2], Yoav Hazan [1] & Amir Rosenthal [1✉]

Optical imaging is commonly performed with either a camera and wide-field illumination or with a single detector and a scanning collimated beam; unfortunately, these options do not exist at all wavelengths. Single-pixel imaging offers an alternative that can be performed with a single detector and wide-field illumination, potentially enabling imaging applications in which the detection and illumination technologies are immature. However, single-pixel imaging currently suffers from low imaging rates owing to its reliance on configurable spatial light modulators, generally limited to 22 kHz rates. We develop an approach for rapid single-pixel imaging which relies on cyclic patterns coded onto a spinning mask and demonstrate it for in vivo imaging of *C. elegans* worms. Spatial modulation rates of up to 2.4 MHz, imaging rates of up to 72 fps, and image-reconstruction times of down to 1.5 ms are reported, enabling real-time visualization of dynamic objects.

[1] Technion – Israel Institute of Technology, Haifa, Israel. [2] These authors contributed equally: Evgeny Hahamovich, Sagi Monin. ✉email: amir.r@ee.technion.ac.il

Digital cameras are one of the most widely used tools in optical imaging today. From mobile phones to advanced microscopy techniques, camera technologies, such as charge-couple devices (CCD) and metal-oxide-semiconductor (CMOS), have revolutionized the world of imaging. Yet, in many applications, digital cameras are not an option. Mature camera technology is available only in a small portion of the electromagnetic spectrum and its performance characteristics may limit some applications. For example, time-domain imaging techniques often require response times shorter than 1 ns[1], whereas in techniques such as full-field optical coherence tomography[2] (OCT) the dynamic range has been limited by camera saturation.

When only a single detector is available, imaging may be performed with collimated or focused illumination that is raster-scanned over the imaged object, while continuously monitoring the detector signal. Raster scanning is the favorable approach in numerous applications, including OCT[3], multi-photon fluorescence microscopy[4], and time-of-flight imaging methods[5]. In addition, raster scanning is common in hybrid imaging techniques in which only the excitation is optical, e.g., optoacoustic[6] and photothermal imaging[7]. While raster-scanning offers more flexibility in detection characteristics than camera-based methods, it imposes more strict limitations on the source. Namely, while cameras can operate with wide-field illumination, raster scanning requires sources with high spatial coherence that enables collimation and focusing.

Single-pixel imaging[8] (SPI) offers an alternative to camera-based and raster-scanning techniques, which requires only a single detector and wide-field illumination, potentially enabling imaging in fields in which both the source and detection technologies are limited[9]. In SPI, configurable spatial light modulators (SLMs) are used to code the illumination with a set of spatial patterns and the detection is performed with a single detector. The signal from the detector is recorded for each pattern projected on the object, and the image is formed from the signals by an inversion algorithm. SLM-based SPI has been demonstrated in numerous applications in optics[9], including visible and infrared microscopy[10], 3D imaging[11,12], fluorescence microscopy[13], multi-spectral imaging[14], multi-photon wide-field imaging[15], viewing through scattering media[16] as well as in hybrid optically assisted techniques, such as optoacoustic imaging[17], optical detection of ultrasound[18], and terahertz imaging[19–21].

In addition to its ability to operate with simple sources and detectors, SPI carries the advantage of compatibility with compressed-sensing theory[22,23]: If the coded patterns are properly chosen, an image with $N$ pixels may be reconstructed with fewer-than-$N$ projected pattern. Thus, compressed sensing may facilitate, at least in theory, imaging at higher rates than conventional approaches, which require $N$ measurement for forming an image with $N$ pixels. However, in practice, the benefits of compressed sensing are often marginalized by the slow refresh rate of most SLMs, which cannot compete with the parallel data acquisition of cameras or with the speed of beam-scanning techniques. Specifically, the most common SLMs in the field of SPI are digital micromirror devices (DMDs), which operate at typical rates of up to 22 kHz—insufficient for dynamic imaging scenarios.

Recently, Xu et al. have demonstrated an alternative approach to SPI in which the structured-illumination patterns are created by a matrix of light-emitting diodes (LEDs), rather than by using SLMs[24]. By using fast-switching LEDs, rates as high as 500 kHz have been demonstrated. While Xu's approach employs only a single-pixel detector and relies on the mathematical principles of SPI for image formation, it deviates from the SPI paradigm due to its reliance on a multi-pixel source. This difference is not merely semantic. A major advantage of SPI is its general compatibility with any optical source, and this advantage is lost in schemes based on multi-pixel sources. In the case of complex sources, e.g., femtosecond lasers used in multi-photon microscopy[15], manufacturing a matrix of individually controlled sources is impractical. Furthermore, in the case of spatially incoherent sources, e.g., LEDs, imaging at microscopic resolutions would require using micron-scale source pixels due to fundamental limitations in focusing spatially low-coherent light. Such dense arrays, if produced with high switching rates and luminance, would involve challenges in both manufacturability and heat dissipation. In the specific implementation of ref. [24], an imaging resolution of 270 μm was reported for images with $32 \times 32$ pixels.

In this work, we show a method for ultra-fast SPI and demonstrate it with rates up to 2.4 million pixels per second achieved without compressed-sensing algorithms. Our proposed scheme is based on illuminating a photomask coded with a predetermined binary pattern to spatially modulate the light. Switching between the patterns is performed by rotating the mask. While mechanically switching between different codes would generally lead to low imaging rates[25], our scheme is based on cyclic codes, in which translation by a single cell leads to a new pattern[26]. Using cyclic codes, $N$ basis functions of a complete, well-conditioned basis can be produced by $N$ single-cell shifts (additional details found in Supplementary Notes 2 and 8). By shrinking the cell size to the scale of the wavelength and employing a fast rotation stage, spatial modulation at megahertz rates is demonstrated for images with up to 25,111 pixels and image-pixel size down to 2 μm. The cyclicity of the basis functions is exploited not only in the experimental setup, but also in the reconstruction algorithms, enabling numerically efficient reconstructions with real-time performance.

## Results

**System configuration**. An illustration of our experimental setup is shown in Fig. 1. The binary codes, consisting of 1's and 0's, were coded in circular pattern on a 6.35-mm thick chrome-coated photomask (Toppan Photomasks). A small portion of the mask, corresponding to a single basis function, was illuminated with a LED operating at 625 nm. The resulting spatial light pattern at the output of the mask was projected onto the imaged sample using objective and tube lenses with a magnification of ×10. The light transmitted through the imaged sample was focused onto a single silicon photodiode. The performance of several masks was tested, with both square and hexagonal cell shapes, with cell widths ranging from 1.5 to 5.2 μm. The photomasks were rotated at a constant speed of 540–600 revolutions per minute. For patterns fabricated 55 mm from the center of rotation and a cell size of 1.5 μm, the transition time between two basis functions, denoted by $T$, was ~0.43 μs. Since the center of rotation did not fully coincide with the center of the circular patterns, the illumination beam was scanned in synchronization with the mask rotation to minimize the radial drift of the projected pattern. The full details of the experimental setup, arrangement of the mask geometries, conversion of 1D code elements into 2D patterns, radial drift compensation, and the variation in cell size due to the changing radius across the pattern are discussed in Supplementary Notes 1, 2, and 3. Supplementary Note 4 discusses additional feedback methods that may be used in case of system stability issues.

**Sampling procedure**. Figure 2 shows an example of the measured signal obtained from a resolution mask and an illustration of the reconstruction procedure. To efficiently transform $N$ distinct

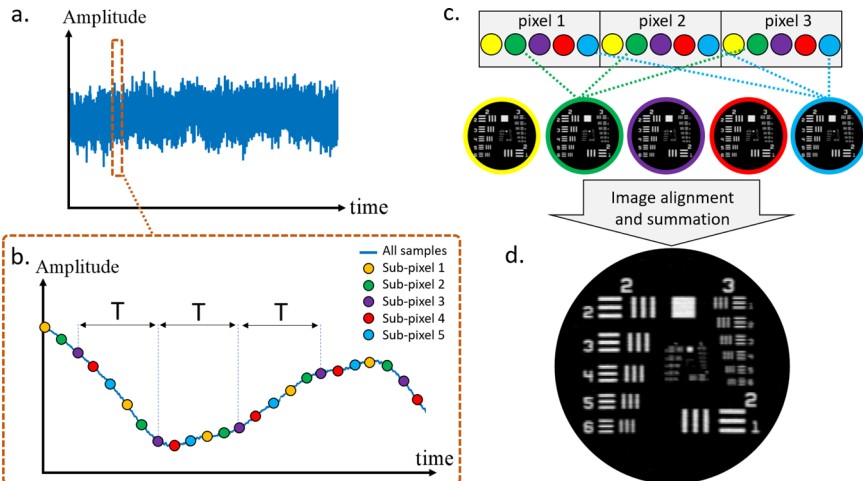

**Fig. 1 Illustration of the rapid-SPI method. a** Conceptual design of the imaging system with the rotating spatially coding mask. **b** Example of a 19th order cyclic mask changed by an element size shift of the mask the colors represent two different base functions. **c** Compatible 19 × 19 cyclic S-matrix.

**Fig. 2 Image formation. a** Full signal sampled from the photodetector. **b** Magnification of the sampled signal, illustrating over-sampling by a factor of $M = 5$, i.e., during the transition time between neighboring basis functions, $T$, which corresponds to a shift of a single pixel between the mask and imaged object, $M$ samples are acquired. Sampled points separated by $T$ appear in the same color in figure (**c**). For a mask with $N$ basis functions, one obtains $M$ datasets with $N$ sampled points each. Each dataset is used to produce its own version of the image by using an FFT-based reconstruction algorithm, where the $M$ reconstructions are identical up to sub-pixel shifts. **d** The final image is obtained by applying sub-pixel shifts to align the $M$ reconstructions and summing the aligned images.

samples of the measured signal into an image with $N$ pixels, we exploit the cyclic property of the sampling matrix and the fast Fourier transform (FFT) (Supplementary Note 5), leading to an image-reconstruction complexity of $O(N\log N)$. Since the model matrix is well-conditioned, no regularization is used in the reconstruction procedure. To fully optimize the signal-to-noise ratio (SNR) of the measurement, the entire signal during the pattern transition time $T$ is used, rather than merely a single discrete sample. Since the transition between neighboring basis functions is gradual, rather than abrupt, we divide the signal during $T$ into $M$ discrete samples, each corresponding to a sub-pixel shift between the mask and imaged object (Fig. 2b). Thus, we obtain $M$ sets of $N$ samples, where each of these $M$ sets is used to reconstruct the $N$-pixel image (Fig. 2c). Since the $M$ resulting images are not identical, but rather represent slightly shifted versions of the imaged object, the images are first aligned to each other via sub-pixel shifts and then summed to form a single image. The benefit of summing sub-pixel-shifted images in terms of SNR and image contrast is demonstrated in Supplementary Note 7.

**Static images**. Figure 3 shows representative SPI images obtained by our system under various experimental conditions with imaging rates as high as 2.4 megapixel/s (the experimental parameters are specified in Supplementary Note 2). Figure 3a–c shows binary images of a resolution target (3a and 3b) and the Technion logo (3c), in which the white regions correspond to full transparency of the imaged objects. To demonstrate the ability of our scheme to produce gray-scale images, we imaged objects that partially absorb or scatter light. For the partially absorbing object, we chose an old photographic film on which a complex image of flowers was captured (Fig. 3d), where for the partially scattering object, a live *C. elegans* worm was used (Fig. 3e). In both cases, higher levels of absorption or scattering in a given region led to darker pixels in the SPI since less light from that region reached the photodetector.

**Dynamic videos**. We demonstrated the capability of our system for dynamic imaging in two distinct scenarios. In the first scenario, the imaged object was a resolution target that was manually

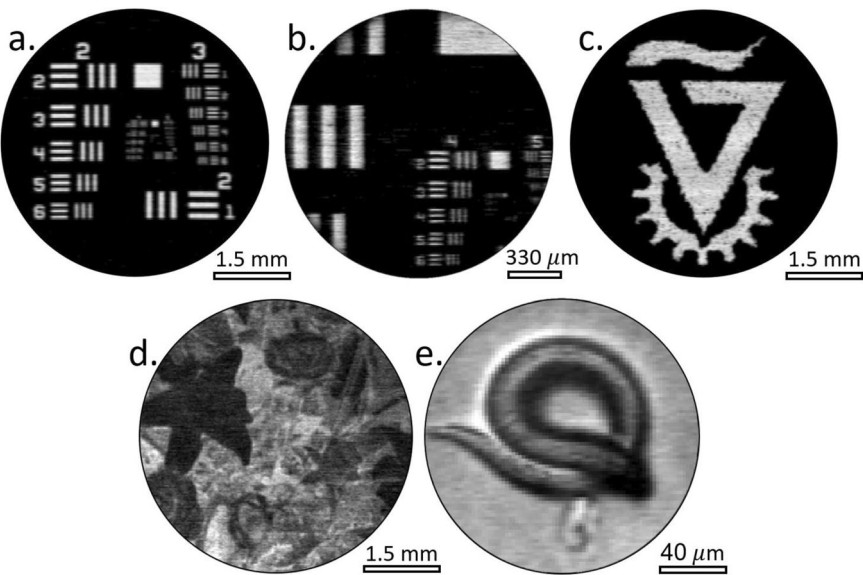

**Fig. 3 Experimental results captured by our system for several mask configurations.** Images (**a** and **b**) are of a resolution target, **c** is a slide of the Technion logo, **d** is an image of an old photographic film with a photo of flowers (comparison to a camera reference is given in Supplementary Note 6), and (**e**) is a microscopy image of a *C. elegans* worm. Images (**a–d**) have 25k pixels and image (**e**) has 10k pixels. Additional configuration details are found in Supplementary Note 2.

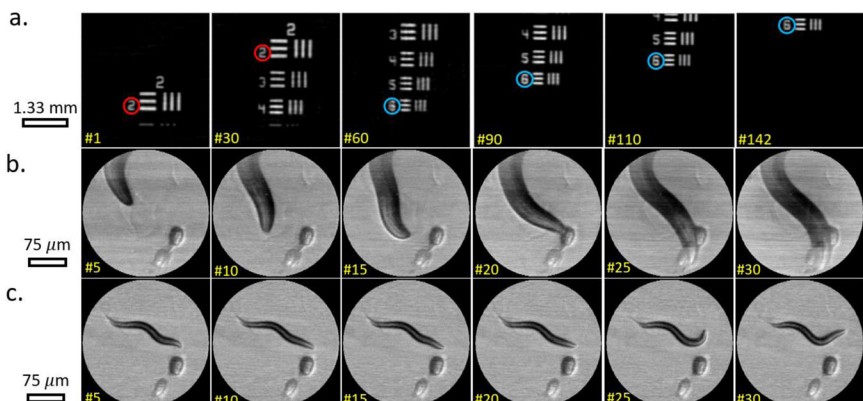

**Fig. 4 Video recording of moving objects. a** Video capturing of vertically shifted resolution target, 101 × 103 resolution and 72 fps frame rate. Total recording rate of 0.75 M pixels per second. Frames 1, 30, 60, 90, 110, and 142 out of the 142 captured frames are presented. The red and blue circles mark constant positions on the resolution target. **b**, **c** Videos capturing the motion of *C. elegans* worms at a frame rate of 10 fps, corresponding to a total recording rate of 0.7 M pixels per second. Frames 5, 10, 15, 20, 25, and 30 out of 31 frames are presented.

scanned in the vertical direction. Imaging was performed with rectangular illumination patterns with $101 \times 103$ ($N = 10{,}403$) pixels, where the imaging speed was 72 frames per second (fps). In the second scenario, freely moving *C. elegans* worms[27] were imaged at 10 fps with hexagonal illumination patterns, where the imaged region was approximately circular and contained $N = 25{,}111$ pixels. The reconstruction time for a single image ranged from 1.5 ms (for $N = 10{,}403$) to 4 ms (for $N = 25{,}111$) of a CPU without parallelization, i.e., shorter than the acquisition time. The full captured videos are found in Supplementary Videos 1–3, whereas snapshots of the videos are presented in Fig. 4. Full details on the captured videos are described in Supplementary Note 10.

## Discussion
In conclusion, we have developed an alternative approach to SPI in which a rapid transition between basis functions is performed by rotating a photomask coded with cyclic patterns. Using our

system, imaging rates of up to 2.4 megapixels/s have been achieved—2 orders of magnitude faster than the rates achievable with conventional DMDs—with a high reconstruction efficiency of $O(NlogN)$, enabling real-time performance for both the data acquisition and image reconstruction. Video-rate imaging has been demonstrated with rates up to 72 fps, enabling us to capture the movement of *C. elegans* worms in vivo. Further increase in the imaging rate may be achieved by using compressed-sensing algorithms as demonstrated in Supplementary Note 9, but at the expense of potentially slower image reconstruction, with a typical complexity of $O(N^2)$ even for highly efficient algorithms[28]. To fully exploit the benefits of compressed-sensing theory, it may be beneficial to replace the well-conditioned basis used in this work with a new type of cyclic, or semi-cyclic, basis in which the incoherence is optimized[29].

In contrast to DMD-based systems, our spatial-modulation scheme is not reconfigurable, and the illumination patterns need to be determined before the production of the mask. However, since the width of the patterns is generally less than a millimeter,

tens of patterns may be produced on a single mask, enabling one to switch between different options of image resolution and size. In some cases, the use of photomasks may offer greater flexibility than DMDs, since it enables arbitrary element geometries and sizes that are not configurable on DMDs.

Our rapid modulation scheme and algorithms may be regarded as a new paradigm for SPI, which is not limited by the speed of DMDs, and may enable new applications in fields in which camera technology is limited. In contrast to schemes based on multi-pixel sources[24], our approach is generic and may be generally used with any optical source or generalized to other wave-based imaging fields by producing appropriate masks for those applications, e.g., thin acoustic blockers in the case of ultrasound[30] or printed circuit boards for terahertz imaging[31]. Alternatively, our scheme may be generalized to other fields by using elements that convert from optics to the field of interest. For example, it has been demonstrated that illuminating a highly resistive silicon wafer with spatially modulated light can enable the spatial modulation of a terahertz beam transmitted through the wafer[21]. With recent advancements in terahertz SPI, the speed of such systems is currently limited by the rates of DMDs, and thus may be improved by using our scheme.

Since the imaging speed of our approach is mainly limited by the speed of the scanning apparatus and bandwidth of the detector and acquisition system, it may, in principle, achieve the high imaging rates of raster-scanning techniques. For example, by using acousto-optic modulators, which can perform line scans at a typical rate of 100 MHz[32], and fast detectors with gigahertz bandwidths, imaging at rates of several gigapixels per second may be possible, transforming SPI into a high-performance method for dynamic imaging.

## Methods

**Statistics and reproducibility**. Each of the captured images a–d presented in Fig. 3 was measured independently for 10 times to confirm results reproducibility. Each one of the captured signals yielded the same result image. The results presented in the paper used only a single capture to perform image reconstruction. Figure 3e was captured only once due to the motion of the imaged worm.

**Reporting summary**. Further information on research design is available in the Nature Research Reporting Summary linked to this article.

## Data availability
The data of the current study is available through zenodo[33].

## Code availability
The code of the current study is available through zenodo[34].

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

## Acknowledgements
We acknowledge the partial financial support provided by the Ollendorff Minerva Center, Prof. Shay Stern for providing us the imaged *C. elegans* samples and Roni Monin for her assistance with biological samples handling.

## Author contributions
E.H. and S.M. designed and constructed the system, performed the experiments, and analyzed the data. Y.H. assisted with the optical setup and A.R. conceived and supervised the project.

## Competing interests
The authors declare no competing interests.
