## [Peer Review File · Nature Communications]

REVIEWER COMMENTS

Reviewer 1:

This manuscript reported a novel modulation method for single-pixel imaging, to increase its imaging speed. Specifically, it replaces the conventional DMD with a rotating mask, which provides different modulation patterns once the rotation proceeds with one cell size. Although the manuscript is technically sound, by only reporting a new modulation strategy does not meet the innovation criteria of Nature Communications. Besides, the following concerns should be addressed.

1. The cyclic nature of the rotating modulation patterns results in different cell sizes for different radius. How to compensate for the different cell sizes and the resulting resolution?
2. Fig. 2 claims that using subpixel integration can improve imaging resolution, which however can not be observed in Fig. 2(c) and (d). The close-ups of resolution comparison should be provided.
3. Although the supplementary material 3 describes the calibration of rotation aberration, the rotation speed may still maintain random disturbance due to unstable voltage. How to compensate for this kind of noise when it is random and hard to calibrate?
4. To validate the dynamic imaging ability of the reported technique, it is better to implement an experiment observing natural and meaningful phenomena, instead of imaging a static object driven by a translation stage.

Reviewer 2:

The paper deals with single pixel imaging approach that allows to significantly increase the rate of the reconstructed images by realizing rapid spatial light modulation which is based on cyclic patterns coded onto a spinning mask, and report spatial modulation rates of up to 2.4 MHz with imaging rates of up to 72 fps. This is without doubt more than two orders of magnitude faster than the state of art existing today in single pixel imaging configurations in the visible range. Experimental results showing high-quality super-resolved reconstruction are presented and they demonstrate the applicability of the proposed concept. Without doubt the paper is very interesting and shows a promising and applicable approach for imaging dynamic objects. I would recommend accepting the paper for publication after revision in which the authors will address the following concerns:

- The demonstrated image has 25K pixels. This is relatively low spatial resolution. Today there are fast cameras (not single pixel) reaching frame rate of much more than 72 fps with resolution of more than 25K pixels. The authors need to relate better on the advantage their concept is bringing in respect to the competition that existing/available fast cameras pose. For instance, a commercial Pixellink camera (a low end and very low cost camera) can provide resolution of about 10K pixels at rates which are more than an order of magnitude faster than those obtained in the proposed concept. High end and expensive cameras give much stronger competition by providing frame rates of about 1M fps (although at lower resolution than 25K pixels).
- The authors don't sufficiently relate to the computation complexity involved in the decoding process. They should address this and relate to the applicability of such complexity for real time images acquisition.
- The experimental results show very nice reconstruction but of only binary images. It would be interesting to see the capability of the proposed approach in dealing with gray level images. This will test the strength of the applied codes and the decoding process in real time scenario.

Reviewer 3:

In the manuscript titled "A new spin on single-pixel imaging: Rapid spatial light modulation via cyclic codes" by Evgeny Hahamovich, Sagi Monin, Yoav Hazan and Amir Rosenthal create a single-pixel imaging setup that overcomes the modulation rate limitation of digital micromirror devices (DMD). They do this by using a cyclic sampling basis, where each projected mask is the previous mask shifted by one pixel. They then put this cyclic sampling basis on a spinning disc which is used as the spatial light modulator. This spinning disc modulator is able to achieve a modulation rate of 2.4MHz which is 2 orders of magnitude quicker than DMD modulation rates.

I think this demonstration deserves attention as the idea does offers very rapid modulation rates with current technology. However, I can not recommend publication for the following reasons:

1. The explanation of sub-pixel shift issue and figure 2 are substandard. Whilst explaining the problem of the continuous shift is explained well, how exactly they have solved it is confusing. In particular, figure 2C looks like they have averaged 5 of the same images together, you are not able to see anything about the sub-pixel shifting issue they have described. Then why is the value changing for each period? Is this the data from their video?

2. They have not fully utilized the properties of cyclic matrices in their image reconstruction. Notably, they have a vector which they shift k -times and then multiply by their k -th measurement. This results in a reconstruction algorithm of complexity N^2 for an N pixel image. However, with a cyclic sampling matrix one can recover their image as follows: Consider the equation $y=Cx$ where y is their measurement, x is the image and C is their cyclic sampling matrix where the k -th row is obtained shifting vector c $(k-1)$ -times. This can expressed as $y=c*x$ where $*$ denotes the circular convolution. Then by the circular convolution theorem one can express this equation in terms of Fast Fourier Transforms: $F\{y\} = F\{c*x\} = F\{c\}F\{x\}$, where $F\{y\}$ denotes the Fast Fourier Transform of y . Then $x=IF\{ F\{y\}/F\{c\} \}$ where $IF\{\}$ denotes the Inverse Fast Fourier Transform. See Chapter 1 (equations 1.80 to 1.85) of <https://doi.org/10.1016/B978-0-08-050780-4.50006-0> for mathematical details. This creates an image recovery algorithm of complexity $N \log(N)$ hence this will be much quicker, and negate the need for CPU parallelization.

They should also mention that their cyclic matrices are related to the Paley construction of Hadamard matrices, just to give new readers extra material they can look up.

3. The compressed sensing aspect of the single-pixel imaging scheme is neglected. The authors have added some images from undersampled data in the supplementary, but this is not enough. In particular this is one of the main disadvantages of their system: they are not able to freely choose the projected masks like DMDs. For example, is there a difference in the reconstructed image if they sample every odd numbered mask and sampled the first 50% of the masks? I suspect this will have a difference, and perhaps not in their favour (ie. sampling every odd numbered mask might give better results), which needs to be discussed so that mathematicians can think about what cyclic masks to create for optimal undersampling in such a spinning disk system.

They also have not plotted any parameters that are normally used to show the effectiveness of various undersampling algorithms, for example the structural similarity index measure or the peak signal-to-noise ratio, as a function of compression ratio. Finally, the work of [K. M. Czajkowski, A. Pastuszczak, and R. Kotyński "Real-time single-pixel video imaging with Fourier domain regularization" Optics Express Vol. 26, Issue 16, pp. 20009-20022 (2018)] needs to be cited as they create an inverse matrix that reconstructs images with quality comparable to the TVAL3 algorithm but with real-time reconstruction via matrix multiplication.

Thus I think expanding the compressed sensing aspect in this works is needed and the moving it in the main part of the manuscript and putting the sub-pixel shifting explanations in the supplementary is needed.

Author Reply to reviewer #1

This manuscript reported a novel modulation method for single-pixel imaging, to increase its imaging speed. Specifically, it replaces the conventional DMD with a rotating mask, which provides different modulation patterns once the rotation proceeds with one cell size. Although the manuscript is technically sound, by only reporting a new modulation strategy does not meet the innovation criteria of Nature Communications.

We thank the reviewer for the detailed comments that helped us improve our paper. Nonetheless, we strongly disagree with the reviewer's statement that our new modulation strategy does not meet the innovation criteria of Nature Communications, especially in light of related papers published in this prestigious journal. Following the reviewers' comments, we have now added additional, new results, which further highlight the innovation of our work. As we explain in the following, we did not merely develop a new modulation strategy, but rather created a completely new, DMD-free, paradigm for SPI, which is now supported with unprecedented results of high frame rate single-pixel imaging of a dynamic organism achieved in the last couple of months.

The concept of single-pixel imaging (SPI) has been in the spotlight of the imaging community for over a decade now, and has led to very innovative imaging devices published in numerous high-impact papers, including papers in Nature Communications. Currently, the field has been limited by the speed of DMD technology. When it comes to imaging, and biomedical imaging in particular, the slow image acquisition of SPI is a fundamental limitation, and not just a technical one. Transforming SPI from a technique that can image only static objects into a technique that can image dynamic objects is a fundamental advancement of the field. Further enhancing our innovation is that we have developed a completely new approach to the spatial modulation, which is based on the mathematics of circulant S-matrices, the fabrication accuracy of photomasks, and a highly accurate beam tracking scheme that enabled us to keep the illumination on the correct part of the photomask with sub-micron accuracy. In contrast, the main approach for enhancing imaging speed until today has been through exploitation of image sparsity, system modelling, and the application of compressed-sensing algorithms. For example, in a very recent Nature Communications paper from May 2020 of Stantchev *et al.* (Ref. 21 in the revised manuscript), the authors managed to reach 6 fps for 32×32 images through proper modelling of the temporal response of the spatial THz modulator and the application of compressed-sensing algorithms.

Thanks to the reviewers' comments, our manuscript includes two more innovations:

- 1) Real-time image reconstruction: While conventional SPI systems generally rely on regularized inversion for image reconstruction, leading to lengthy reconstruction times, the use of circulant basis enables real-time reconstruction with an efficiency of $O(N \log N)$, enabling us to reconstruct an image in 4 ms on a standard CPU with no parallelization.
- 2) In vivo imaging: In the revised version, we imaged the movement of *C. elegans* worms, a widely used model organism in developmental biology. This is the first time that SPI was demonstrated for imaging the movement of a live organism in high detail, further illustrating the potential impact of our work.

In summary, we offer a completely new approach to SPI (novel circulant codes + accurate mechanical rotation of a photomask) with new reconstruction algorithms that bridges the gap between this highly

researched imaging approach and biomedical applications, in which rapid imaging is often required. In terms of results, we have now demonstrated the first *in vivo* application of video-rate SPI, which was also obtained with real-time image reconstruction. Our approach may be considered as a new paradigm in SPI. In the past, the focus has been reducing the number of samples, which might have increased the data acquisition speed, but increased the computational burden, effectively increasing the total time required to form an image. In our approach, the choice of a circulant basis enabled us to perform both the data acquisition and the reconstruction in real time, making SPI a truly real-time method.

If we take the work of Stantchev *et al.*, recently published in Nature Communications, as a measure for the innovation level required to publish in this journal, we find it difficult to argue that our work does not reach the same level of innovation, if not surpasses it. In that work, the experimental setup was standard: a DMD was used to spatially modulate a laser beam, which illuminated a high resistivity silicon layer, spatially modulating its THz transmissivity. Most of the innovation was in the modelling and the unprecedented speeds achieved for single-pixel THz imaging. In our work the innovation is first and foremost conceptual, offering a new paradigm in which the mathematical properties of circulant bases can enable rapid spatial modulation via mechanical scanning (or alternatively beam steering as discussed in the last paragraph), thus avoiding the need for using DMDs, which have been the corner-stone of almost any SPI system for over a decade. Then, we offer the experimental innovation of the spinning-mask system with beam tracking that prevented a drift of the illumination pattern. We further offer the innovation of ultra-fast image reconstruction for real-time image visualization, and finally our results go well beyond all previous attempts for video-rate SPI, enabling, for the first time, imaging of moving organisms.

Besides, the following concerns should be addressed.

1. The cyclic nature of the rotating modulation patterns results in different cell sizes for different radius. How to compensate for the different cell sizes and the resulting resolution?

Indeed, some difference exists between the width of the elements placed on the internal and external ring. However, this difference is marginal in comparison to other parameters in the system. The following example compares the element size of an internal and external elements of the coding pattern for Fig. 4a of the main paper. This calculation is now included in section 2 of the Supplementary Material.

For the images presented in Fig. 4a of the paper, the aperture size is $404\mu\text{m} \times 412\mu\text{m}$, and the pattern is located on a 57.2mm radius. Therefore, the difference between the inner and the outer element sizes is $0.03\mu\text{m}$. This 0.75% change in the element size is well below the lithographic design resolution achieved by our manufacturing procedure and is comparable to the critical-dimension uniformity of the mask. Accordingly, the effect of radius variations within the pattern may be neglected.

Fig. 1 of this rebuttal letter shows the detailed geometry, followed by detailed calculations.

Review response Fig. 1 | Pattern variation due to the circular geometry of pattern aperture. Example calculation for Fig.4a of the paper. Pattern distance of 57.2 mm and element size of $4\mu\text{m}$ results in difference of 30 nm in element size.

Supporting calculations:

$$\text{Angle: } \theta = \arctan\left(\frac{412}{57200}\right) = 0.0072$$

$$\text{Total width for external radius: } 57402 \cdot \tan(0.0072) = 413.89\mu\text{m}$$

$$\text{Total width for internal radius: } 56998 \cdot \tan(0.0072) = 410.4\mu\text{m}$$

$$\text{Total difference is } 3.4\mu\text{m}, \text{ and element difference is } \frac{3.4}{103} = 0.03\mu\text{m}$$

2. Fig. 2 claims that using subpixel integration can improve imaging resolution, which however can not be observed in Fig. 2(c) and (d). The close-ups of resolution comparison should be provided.

We thank the reviewer for the remark.

Indeed, as was previously mentioned in our concluding paragraph “the availability of subpixel-shifted images can enable algorithmic increase of image resolution”. This was a general statement about previous works in the field, in which sub-pixel shifts were used for super-resolution. However, the reviewer is correct that we have not implemented any of these methods in our own work, and therefore this statement may lead to confusion. We have therefore omitted this comment from the manuscript. The text discussing Fig. 2 has also been modified and now better explains that the reason for using the signals obtained from sub-pixel shifts is to optimize the SNR. There is no resolution enhancement in the transition between Fig.2(c) and Fig. 2(d), only enhanced SNR due to the averaging effect. Nonetheless, the sub-pixel realignment is performed to reduce the resolution degradation that would have occurred if the slightly shifted images were summed directly.

To showcase the effect of the realignment on image contrast and resolution, we compared its resulting reconstruction to the case in which it is not applied (Section 7 in the Supplementary Information). The comparison is also shown in Fig.2 of this rebuttal letter. As can be seen in the reconstructions of Targets 3.1 and 3.2, realigning the images enhances the contrast of fine details in the image.

Review response Fig. 2 | Comparison of sub-pixel shifted averaging to standard averaging. $M = 5$ slightly shifted experimental reconstructions of a resolution target were produced and summed to produce the final reconstruction (a) with and (b) without performing sub-pixel realignment before summation. (c)-(f) a comparison between the 1D slices of resolution targets 2.5, 2.6, 3.1 and 3.2, reconstruction with (red) and without (blue) realignment.

3. Although the supplementary material 3 describes the calibration of rotation aberration, the rotation speed may still maintain random disturbance due to unstable voltage. How to compensate for this kind of noise when it is random and hard to calibrate?

We fully agree that random noise in the velocity can, in theory, lead to fluctuations in the mask position, which could not be compensated by a calibration procedure that was performed *a priori*.

Indeed, when we originally designed the system, we had concerns that major velocity fluctuations would severely limit our performance. We therefore developed a technique to monitor the mask position and velocity during its rotation, which we have now added as section 4 of the Supplementary Materials. We used this monitoring technique to perform preliminary tests in which we assessed the rotation velocity under various conditions. Our conclusion was that the rotation speed of the stage was consistent. Generally, our concern was more with slow variations in velocity, rather than the effect of high-frequency noise in the voltage supply of the stage since the momentum of the stage and mask effectively flattened any spikes in the acceleration of the stage. Since our conclusion was that the stage rotation speed was consistent, we did not perform the monitoring during the imaging sessions – a procedure that would have required adding more components to the system. Indeed, our good imaging results are an indicator that velocity monitoring was not essential in our specific experimental setup. Nonetheless, as we explain in the following, velocity monitoring during imaging is possible using our scheme, and can enable more accurate modeling or the use of less stable stages.

To measure the rotation speed of the stage, we fabricated a periodic pattern on the mask, and illuminated it instead of the coded pattern used for imaging. Our periodic pattern was composed of segmented lines with 50% duty-cycle, as presented in Fig. 3 of this letter. A light source was focused

to a small point illuminating the segmented line circumference at the antipodal point of the main illumination. The light transmitted through the segmented line was measured by a photodiode, leading to a signal that alternated between the two states. Denoting the length of the segments by x , and the alternation time between two states with t_i , the velocity was calculated using $v_i = x/t_i$. Fig. 4 of this letter shows an example experiment performed. Figs 4a and 4b show an example of the photodiode signal obtained during the acceleration of the stage and Fig. 4c show the velocity, extracted from the performed measurement. As now discussed in section 4 of the Supplementary Materials, the basic scheme for velocity monitoring can be integrated in the imaging scheme by simultaneously illuminating both the coded pattern used for imaging and the periodic pattern used for monitoring, where the light transmitted through each pattern would be collected separately.

Review response Fig. 3 | Segmented lines with 50% duty cycle printed on circumference of the mask. The image was captured with camera.

The technique described above may be used to monitor the rotation speed of the mask during the imaging session by adding a second photodetector to capture the transmission pattern from the periodic pattern. Then, drifts in velocity may be corrected during data post-processing.

Review response Fig. 4 | Velocity measurement using the segmented line pattern. (a) voltage from the photodetector recorded during stage acceleration, (b) zoom on the captured pattern and (c) the recovered velocity of the stage as a function of time.

4. To validate the dynamic imaging ability of the reported technique, it is better to implement an experiment observing natural and meaningful phenomena, instead of imaging a static object driven by a translation stage.

We conducted additional experiments with *C. elegans* worms and additional results now appear in both the manuscript and the Supplementary Materials. This is the first time that SPI was demonstrated for imaging the movement of a live organism, further illustrating potential impact of our work. We note that while our masks were designed for continual image acquisition (i.e. they formed a complete circle and the object was continuously illuminated), the finite memory of our oscilloscope limited the duration of our videos to 30-32 frames. Since the number of total frames was limited, in the results presented in the main document we chose to operate at a frame rate of 10 fps to capture more of the worm movement. Nonetheless, real-time data acquisition systems, in which the data is directly transmitted to the computer (enabling effectively unlimited imaging durations), are commonly used in raster-scanning imaging systems, e.g. OCT and mutli-photon fluorescence microscopy, and can reach data-transmission rates far higher than those demonstrated in the current work.

Author Reply to reviewer #2

The paper deals with single pixel imaging approach that allows to significantly increase the rate of the reconstructed images by realizing rapid spatial light modulation which is based on cyclic patterns coded onto a spinning mask, and report spatial modulation rates of up to 2.4 MHz with imaging rates of up to 72 fps. This is without doubt more than two orders of magnitude faster than the state of art existing today in single pixel imaging configurations in the visible range.

Experimental results showing high-quality super-resolved reconstruction are presented and they demonstrate the applicability of the proposed concept. Without doubt the paper is very interesting and shows a promising and applicable approach for imaging dynamic objects. I would recommend accepting the paper for publication after revision in which the authors will address the following concerns:

We thank the reviewer for their support and useful comments, which helped us improve the quality of our work. In particular, the reviewer's suggestion to include non-binary images has helped us to showcase the potential of our scheme for high-end imaging applications.

The demonstrated image has 25K pixels. This is relatively low spatial resolution. Today there are fast cameras (not single pixel) reaching frame rate of much more than 72 fps with resolution of more than 25K pixels. The authors need to relate better on the advantage their concept is bringing in respect to the competition that existing/available fast cameras pose. For instance, a commercial PixelLink camera (a low end and very low cost camera) can provide resolution of about 10K pixels at rates which are more than an order of magnitude faster than those obtained in the proposed concept. High end and expensive cameras give much stronger competition by providing frame rates of about 1M fps (although at lower resolution than 25K pixels).

We fully agree with the reviewer that commercial camera technology is vastly superior to SPI when it comes to conventional imaging (photography or microscopy in the visible spectrum). Indeed, the motivation to our work, and the general justification for SPI, is not as an alternative to mature camera technologies, but rather as an imaging approach in fields in which cameras technology is inadequate. As stated in the introduction "Mature camera technology is available only in a small portion of the electromagnetic spectrum and its performance characteristics may limit some applications. For example, time-domain imaging techniques often require response time shorter than 1 ns, whereas in techniques such as full-field optical coherence tomography (OCT) the dynamic range has been limited by camera saturation."

While unattractive for photography, SPI has shown promise for advancing imaging fields that currently involve complex instrumentation and phenomena. However, the current speed limitation of DMDs may limit such applications to proof-of-concept studies and prevent their use for imaging dynamic objects. Specific examples include:

1. Hybrid imaging in Terahertz wavelengths. THz camera technology today is expensive and limited (see discussion in Ref. 20 in the revised manuscript) and lacks high-resolution spectroscopic capabilities that SPI systems possess. The most advanced form of THz SPI is based on spatially modulated illumination patterns (produced using DMDs) that change the transmissivity of semiconductors, effectively generating a THz mask (e.g. reference 19-21 in the revised manuscript). With the very recent advancement of Stantchev *et al.* (Ref. 21 in the revised manuscript), the imaging rate of SPI-based THz is now limited by the speed of DMDs, rather than

by the response time of THz components. Our work can thus enable the next generation of SPI-THz imagers.

2. Hybrid imaging in opto-acoustic imaging, where ultrasonic detectors are based on optic-detectors such as Fabry-Perot detector (Reference 17 and 18 in the revised manuscript).
3. Multi-photon imaging in scattering medium using temporal focusing. It has been recently shown by Adrià Escobet-Montalbán et al. (Ref. 15 in our revised manuscript) that SPI can improve the performance of 2p-temporal-focusing microscopes over using a camera. The main reason for this result was the use of ballistic photons only for illumination, whereas the camera-based system required ballistic propagation also for detection.

While much of the above text was already expressed in the introduction of the original manuscript, we have now added the example of multi-photon imaging and addressed the recent development in SPI THz imaging, in addition to adding a short discussion in the concluding paragraphs.

The authors don't sufficiently relate to the computation complexity involved in the decoding process. They should address this and relate to the applicability of such complexity for real time images acquisition.

In the revised version of our manuscript we discuss the issue of complexity and the capabilities of our method not only for real-time data acquisition, but also image formation, and present a new reconstruction algorithm with superior efficiency. In the original manuscript, the reconstruction was performed by a matrix-vector multiplication, leading to a reconstruction complexity of $O(N^2)$ and reported reconstruction times of 0.5 s on a CPU without parallelization, considerably more efficient than most compressed sensing algorithms, which rely on iterative optimization algorithms. Since the performance achieved in the original manuscript were obtained with a CPU, they already had the potential for video-rate reconstructions since matrix-vector products can be calculated much faster with on GPUs.

In the current version of our manuscript, we have implemented a new reconstruction algorithm, following the suggestion of Reviewer 3. By exploiting the circular nature of the basis and using the fast Fourier transform in the reconstruction, we have now reached a complexity of $O(N \log N)$, reducing reconstruction time to 4 ms on a CPU. Thus, our current reconstruction time is now significantly shorter than the acquisition time (our maximum speed was 72 fps corresponding to 14 ms acquisition time for a single image), a result we achieved without any parallelization. The details of the fast reconstruction algorithm now appear in section 5 of the Supplementary Materials.

As suggested by the reviewer, we now highlight that our scheme does not only enable video-rate data acquisition, but also video-rate image reconstruction, enabling us to perform the entire imaging procedure in real time. Indeed, real-time visual feedback is important in numerous imaging applications.

The experimental results show very nice reconstruction but of only binary images. It would be interesting to see the capability of the proposed approach in dealing with gray level images. This will test the strength of the applied codes and the decoding process in real time scenario.

As suggested by the reviewer, we used our system to image a slide of an old photographic film from a film camera to present gray levels handling by our technique. The image is shown in Fig. 1 of this response in comparison to the same object imaged with regular camera and is also shown in section 6 of the Supplementary materials. Additionally, to present the real time scenario, we have tested our system's capability of imaging live organisms. Specifically, we imaged the movement of *C. elegans* worms (Videos 2 and 3 in the supplementary materials and Fig. 2 in this rebuttal letter). These images clearly show that our system is capable of capturing texture and the visual quality of our images is on par with, if not superior to, the most advanced experimental implementations of SPI.

Review response Fig. 1 | Gray level image of a flowers slide. An image of an old photographic slide with gray levels captured with our SPI system (left) is compared to a reference image (right) taken with a camera.

Review response Fig. 2 | High frame rate imaging of *C. elegans* worms. The figure shows several snapshots taken from two captured SPI videos in which the movement of a *C. elegans* worms was recorded.

Author Reply to reviewer #3

In the manuscript titled "A new spin on single-pixel imaging: Rapid spatial light modulation via cyclic codes" by Evgeny Hahamovich, Sagi Monin, Yoav Hazan and Amir Rosenthal create a single-pixel imaging setup that overcomes the modulation rate limitation of digital micromirror devices (DMD). They do this by using a cyclic sampling basis, where each projected mask is the previous mask shifted by one pixel. They then put this cyclic sampling basis on a spinning disc which is used as the spatial light modulator. This spinning disc modulator is able to achieve a modulation rate of 2.4MHz which is 2 orders of magnitude quicker than DMD modulation rates.

I think this demonstration deserves attention as the idea does offers very rapid modulation rates with current technology. However, I can not recommend publication for the following reasons:

We thank the reviewer for their support and useful comments, which helped us improve the quality of our work. We are especially grateful for the reviewer's suggestion for an improved reconstruction algorithm, which was truly transformative to our work. We have made substantial changes in our work, and implemented all the suggestions made by the reviewer. Our current manuscript includes two additional accomplishments with respect to the original version:

- 1) Complete high frame rate video imaging: While in the original manuscript the data acquisition was performed at video rate (72 fps), the reconstruction was much slower (up to 0.5s on a CPU). Indeed, our argument was that parallelization with GPU would take also the reconstruction towards video-rate performance. In the current version, by implementing the reviewer's excellent suggestion, we improved the reconstruction time by over 2 orders of magnitude, enabling the entire imaging procedure to operate in real-time without any parallelization. While in many cases, it is sufficient to have only rapid data acquisition while performing the image formation offline, for high-throughput biomedical studies and in applications that require real-time feedback, complete video-rate imaging may be essential.
- 2) In vivo imaging: In the revised version, we imaged the movement of *C. elegans* worms, a widely used model organism in developmental biology. This is the first time that SPI was demonstrated for imaging the movement of a live organism, further illustrating potential impact of our work.

1. The explanation of sub-pixel shift issue and figure 2 are substandard. Whilst explaining the problem of the continuous shift is explained well, how exactly they have solved it is confusing. In particular, figure 2C looks like they have averaged 5 of the same images together, you are not able to see anything about the sub-pixel shifting issue they have described. Then why is the value changing for each period? Is this the data from their video?

We substantially modified our explanation how the problem of continuous shift is solved and added a comparison to a naïve reconstruction approach which ignores sub-pixel shifts in section 7 of the Supplementary Information. In conventional DMD-based SPI, the measured signal would be constant over the switching time T and then abruptly switch to the next one. In our case, this transition is smooth, as illustrated in Fig. 2b of the manuscript. The image in this example is static (not video) and the change in signal value is a result of the continuous transition between patterns. One can easily solve this problem by ignoring the data in the transition, and just take samples with sampling intervals

of T . This approach, however, would reduce the SNR of the reconstruction, as much of the data is not used. In our solution, we divide the signal during T into M discrete samples, each corresponding to a sub-pixel shift between the mask and imaged object (Fig. 2b in the manuscript). Thus, we obtain M sets of N samples, where each of these M sets is used to reconstruct the N -pixel image (Fig. 2c in the manuscript). Since the M resulting images are not identical, but rather represent slightly shifted versions of the imaged object, the images are first aligned to each other via sub-pixel shifts and then summed to form a single image.

To showcase the effect of the realignment on image contrast and resolution, we compared its resulting reconstruction to the case in which it is not applied (Section 7 in the Supplementary Information). The comparison is also shown in Fig.1 of this rebuttal letter. As can be seen in the reconstructions of Targets 3.1 and 3.2, realigning the images enhances the contrast of fine details in the image.

Review response Fig. 1 | Comparison of image reconstruction from over-sampled data with and without image realignment. $M = 5$ slightly shifted experimental reconstructions of a resolution target were produced and summed to produce the final reconstruction (a) with and (b) without performing sub-pixel realignment before summation. (c)-(f) a comparison between the 1D slices of resolution targets 2.5, 2.6, 3.1 and 3.2, reconstruction with (red) and without (blue) realignment.

2. They have not fully utilized the properties of cyclic matrices in their image reconstruction. Notably, they have a vector which they shift k -times and then multiply by their k -th measurement. This results in a reconstruction algorithm of complexity N^2 for an N pixel image. However, with a cyclic sampling matrix one can recover their image as follows: Consider the equation $y=Cx$ where y is their measurement, x is the image and C is their cyclic sampling matrix where the k -th row is obtained shifting vector c ($k-1$)-times. This can be expressed as $y=c*x$ where $*$ denotes the circular convolution. Then by the circular convolution theorem one can express this equation in terms of Fast Fourier Transforms: $F\{y\} = F\{c*x\} = F\{c\}F\{x\}$, where $F\{y\}$ denotes the Fast Fourier Transform of y . Then $x=IF\{F\{y\}/F\{c\}\}$ where $IF\{\}$ denotes the Inverse Fast Fourier

Transform. See Chapter 1 (equations 1.80 to 1.85) of <https://doi.org/10.1016/B978-0-08-050780-4.50006-0> for mathematical details. This creates an image recovery algorithm of complexity $N \log(N)$ hence this will be much quicker, and negate the need for CPU parallelization.

They should also mention that their cyclic matrices are related to the Paley construction of Hadamard matrices, just to give new readers extra material they can look up.

We are extremely grateful for this comment, which has not only enabled us to drastically improve our reconstruction times (from 0.5 s to 4 ms), but also serves further justification for the new paradigm we have developed for SPI. Conventionally, to improve the acquisition rate, compressed sensing has been used, which came at the cost of slower reconstruction times. Although reconstruction times may also be improved via GPUs, this does not change the fundamental trade-off of the CS paradigm: faster acquisition at the cost of slower reconstruction. Using cyclic codes (with full, rather than partial, projection of the base) enables the acceleration of both data acquisition and image reconstruction, breaking the trade-off of the CS paradigm. Indeed, the work of K. M. Czajkowski *et al.*, suggested by the Reviewer in their 3rd comment, also breaks the tradeoff of the CS paradigm, but it does not offer the accelerated reconstruction of cyclic bases.

Details of reconstruction now appear in section 5 of the Supplementary Materials.

3. The compressed sensing aspect of the single-pixel imaging scheme is neglected. The authors have added some images from undersampled data in the supplementary, but this is not enough. In particular this is one of the main disadvantages of their system: they are not able to freely choose the projected masks like DMDs. For example, is there a difference in the reconstructed image if they sample every odd numbered mask and sampled the first 50% of the masks? I suspect this will have a difference, and perhaps not in their favour (ie. sampling every odd numbered mask might give better results), which needs to be discussed so that mathematicians can think about what cyclic masks to create for optimal undersampling in such a spinning disk system.

They also have not plotted any parameters that are normally used to show the effectiveness of various undersampling algorithms, for example the structural similarity index measure or the peak signal-to-noise ratio, as a function of compression ratio. Finally, the work of [K. M. Czajkowski, A. Pastuszczak, and R. Kotyński "Real-time single-pixel video imaging with Fourier domain regularization" *Optics Express* Vol. 26, Issue 16, pp. 20009-20022 (2018)] needs to be cited as they create an inverse matrix that reconstructs images with quality comparable to the TVL3 algorithm but with real-time reconstruction via matrix multiplication.

Thus I think expanding the compressed sensing aspect in this work is needed and the moving it in the main part of the manuscript and putting the sub-pixel shifting explanations in the supplementary is needed.

Indeed, additional analysis of the compatibility of our scheme with compressed sensing was in order, and we have made the requested changes, as we explain below. Additionally, we moved some of the analysis of the sub-pixel shifts to the Supplementary Information. Finally, we added a discussion on the potential of using CS with our scheme in the concluding paragraphs. Nonetheless, we strongly feel that the detailed CS analysis belongs in the Supplementary Information, and not in the main text for the following reasons:

- 1) In the past, the slow rates of DMDs have made compressed sensing almost an integral part of previous attempts at video-rate SPI, our aim is to propose a different paradigm in which

compressed sensing is not essential for rapid imaging. With x100 increase in spatial modulations rates, and with even faster modulations possible using other scanning approaches, the importance of CS is diminished. Since our main message is “video-rate SPI *without* compressed sensing”, adding compressed-sensing results to the main text might go against our main message. In other words, we propose that as long as there is no significant improvement in the speed of DMDs, the way towards rapid SPI is not through programmable devices and smart algorithms, but rather through cyclic bases with rapid scanning, where the test should be the imaging rate (pixels/sec) for a given image quality and not the mathematical efficiency in which the data is collected. In that sense, efficient compressed sensing serves only as a partial mean towards a greater end, rather than the main goal.

- 2) The Reviewer’s suggestion of using FFT in the reconstruction further strengthens our proposed paradigm by reducing the computation complexity to $O(N\log N)$, whereas even the efficient approach of Czajkowski *et al.*, which we now cite, had a complexity of $O(NK)$, where K is the number of samples. Accordingly, the highly efficient FFT-based reconstruction algorithm is now an integral part of the proposed paradigm, which is incompatible with CS theory.
- 3) Since our patterns have not been optimized for CS, and indeed are less compatible with it than random patterns, CS is currently not part of proposed paradigm, but rather a future direction that would require more research (either different cyclic patterns or reconstruction approach). As such, we do not think it should be in the main text, but rather in the Supplementary Information.

Still, we fully agree with the Reviewer that our work raises important questions about what cyclic basis is most compatible with compressed sensing, which we now discuss in final paragraphs of the paper. We believe that our work paves a clear path towards ultra-fast SPI based on cyclic masks with even faster scanning (e.g. acousto-optic deflectors) and possibly more sophisticated choice of cyclic bases, designed for CS.

According to the Reviewer’s comment, we now report in section 9 of the Supplementary Information additional results comparing between consecutive samples, under-sampling (as proposed by the reviewer), and random samples.

Review response Fig. 2 | Compressed sensing image restoration based on partial data. The Technion logo image is recovered from partial samples with different sampling procedures.

SSIM	25%	33%	50%	75%
Signal truncation	0.6	0.66	0.82	0.88
Uniform Under-sampling	0.79	0.89	0.93	0.96
Random under-sampling	0.81	0.85	0.9	0.92

PSNR	25%	33%	50%	75%
Signal truncation	12.35	12.44	15.55	18.3
Uniform Under-sampling	18.55	18.59	26.46	27.41
Random under-sampling	16.68	14.98	22.77	25.76

Review response Tab. 1 | SSIM comparison of CS results. Comparison was conducted between CS results and reconstruction with 100% of measurements.

Review response Tab. 2 | PSNR comparison of CS results. Comparison was conducted between CS results and reconstruction with 100% of measurements.

We show the partial compatibility of our system with CS in Fig. 2 of this letter. All reconstruction where done with TVAL3 algorithm with identical hyper-parameters. We show comparison of three types of under-sampling methods:

1. **Signal truncation:** the signal is truncated, leading to a reduced dataset in which samples are consecutive.
2. **Uniform under-sampling:** we take every n^{th} sample of the original dataset, where $n = 2,3,4$, to form the reduced dataset. For 75% we omit every 4th measurement.
3. **Random under-sampling:** the samples of the reduced dataset are randomly chosen from the full dataset.

SSIM and PSNR values are calculated vs image reconstructed with 100% measurements and the results are shown in Tab. 1 and Tab. 2 of this letter.

We see that as expected under-sampling and random measurements achieve better reconstruction with same sampling rates. However, we can see that consecutive measurements with 50% of samples achieve close reconstruction to even/random with 25% of samples.

REVIEWER COMMENTS

Reviewer #2 (Remarks to the Author):

In their revision, the authors have addressed my concerns and I now recommend accepting the manuscript for publication

Reviewer #3 (Remarks to the Author):

The revised manuscript of "A new spin on single-pixel imaging: Rapid spatial light modulation via cyclic codes" is significantly improved and they have adequately answered by previous concerns. However, there is still some problems with the manuscript.

The main problem, which is somewhat related to reviewer 1's comments, is that the manuscript fails to explain the main benefit of this technique and put itself correctly among the existing literature. For example, they do not mention the work [Zi-Hao Xu, et al '1000 fps computational ghost imaging using LED-based structured illumination' Optics Express Vol. 26, Issue 3, pp. 2427-2434 (2018)] where they use an LED matrix array to generate their spatial patterns and they achieve pattern switch rates of 500kHz whilst having much more control over their projected masks. Further, I think the system here will be much more sensitive to outside vibrations than Zi-Hao Xu's. If only comparing this, then this work probably fails to meet the innovation criteria. However, I think it meets the innovation criteria because the Cyclic Hadamard masks used here have by in-large been ignored by the research community and there is no experimental study showing their greatness. This is because mathematicians have been the ones considering the sampling patterns in terms of noise-suppression and undersampling efficiency, but they have not considered the experimental implementation of the masks. These Cyclic masks have the same noise suppression capabilities as ordinary Hadamard masks as well as Fourier masks (which is the best you can do with any sampling matrix due to their orthogonality as shown in ref 25 of the manuscript) and their reconstruction algorithms are of the same complexity but their undersampling efficiency is worse. However, the experimental implementation of these cyclic masks is their greatest asset; and as this work shows one can now easily create a spatial 2D modulator with megahertz switch rates with nearly 100% modulation depth for many different types of waves. For example, for terahertz radiation they can use printed circuit board manufacturing techniques to create their spinning disk or for sound-waves they can just drill some holes. The fact that the number of image pixels is not easily changeable is also true of CCD imaging technology hence it is not a major disadvantage over other spatial light modulators, although it does need to be mentioned.

Until the authors have adequately discussed all the points in the above paragraph in the main manuscript this work will fail to show its greatness and discuss its place relative to other works. As such I can not recommend publication until this is done. Consequently, the manuscript title should be "A new spin on single-pixel imaging: spatial light modulation at megahertz switch-rates via cyclic Hadamard masks" because "cyclic codes" is a field related to cryptography hence creating confusion and "rapid" is ambiguous.

Another issue is that they should say the improvement in signal-to-noise ratio (SNR) from their sub-pixel shifting procedure. For example, from the image in fig 2d they can obtain their SNR by taking the average of the signals transmitted through the large square and then divide it by the root-mean-square of some completely black area. What is the difference in SNR if they only get an image from sub-pixel 3 and then from the average of all 5 sub-pixels (after their alignment procedure). They also need to state their image pixel resolution and not just say $N = 25111$ in the figure captions, which I assume is something around a 50x50 pixel image but it is not exactly.

They need to add scale bars to their videos and other images.

Supplementary figure 1, the tilt/tip mirror looks like it is just a normal mirror. They should add something to make it easily know this mirror is electrically controlled in sync with the rotation of the disc.

Rayko Ivanov Stantchev

Author's Response to Reviewers

Reply to reviewer #2

We thank the reviewer for the insightful feedback during the review process and their positive decision in the current round.

In their revision, the authors have addressed my concerns and I now recommend accepting the manuscript for publication.

Reply to reviewer #3

The revised manuscript of "A new spin on single-pixel imaging: Rapid spatial light modulation via cyclic codes" is significantly improved and they have adequately answered by previous concerns.

We thank the reviewer for the insightful feedback during the review process, which has helped us to significantly improve the quality of our manuscript. We have revised the manuscript based on the comments from the Reviewer and highlighted all changes in an additional version of the manuscript.

However, there is still some problems with the manuscript. The main problem, which is somewhat related to reviewer 1's comments, is that the manuscript fails to explain the main benefit of this technique and put itself correctly among the existing literature. For example, they do not mention the work [Zi-Hao Xu, et al '1000 fps computational ghost imaging using LED-based structured illumination' Optics Express Vol. 26, Issue 3, pp. 2427-2434 (2018)] where they use an LED matrix array to generate their spatial patterns and they achieve pattern switch rates of 500kHz whilst having much more control over their projected masks. Further, I think the system here will be much more sensitive to outside vibrations than Zi-Hao Xu's. If only comparing this, then this work probably fails to meet the innovation criteria.

We fully agree that the important work of Zi-Hao Xu should have been cited and discussed and we have added such a discussion to the main paper. We also agree with the Reviewer's later comment that our ability to turn the mathematical concept of cyclic Hadamard codes into a working imaging system sets us apart from Xu's work and represents a major *conceptual* development in the field on SPI. However, we strongly disagree that our work does not meet the innovation criteria of Nature Communications in terms of new imaging capabilities. As we explain in the following, our scheme offers also *practical* advantages over the work of Xu, and not just conceptual ones:

- 1) The work of Xu is inherently limited to LEDs, or other simple sources, and cannot be readily adapted for complex sources. One of the main advantages of SPI is that it can turn any type of illumination source into an effective multi-pixel source by performing spatial modulation on it. This general property has enabled sophisticated application of the SPI to fields like 2-photon microscopy, in which complex sources are used, namely femtosecond lasers. **But the work of Xu does not involve spatial modulation, but rather an array of illumination sources. Accordingly, Xu's approach can only work with sources that can be produced in dense arrays, which limits the generality of the SPI concept. It is fair to argue that Xu's work is not purely "single pixel" since they replace the multi-pixel camera with a multi-pixel source – a technology that is generally less versatile than cameras.** Using this approach for applications that require complex sources, e.g. 2-photon microscopy or optoacoustics, would be impossible.
- 2) Although the imaging rates of Xu are comparable to ours, we have demonstrated a significantly higher level of performance, owing to fundamental advantages of our scheme:
 - a. Xu's work produced images with a pixel size of approximately 270 μm , whereas our resolution was 2 μm for the *in vivo* images. The reason for that difference is fundamental. LEDs are spatially incoherent and relatively isotropic; therefore, they cannot be focused by large factors without filtering out most of their power. This is a direct result of the fact that the luminance, i.e. optical power per area per solid angle,

is an invariant in geometrical optics: If we want to focus an optical beam to a smaller spot, i.e. increase its power per area, the solid angle covered by its k-vectors must increase. However, LEDs illuminate (almost) in all directions, i.e. their solid-angle coverage is high to begin with; since the solid-angle coverage cannot exceed 4π , only limited focusing can be performed without loss of power.

Accordingly, Xu's scheme is generally incompatible with microscopy. In the implementation of Xu, the focusing was approximately by a factor of 5. In our work, we also used a LED source, but we did not need strong focusing to achieve high resolution because our spatial masks had micron-sized elements. Effectively, our use of masks with micron-sized elements is equivalent to an LED array with micron-sized pixels, with a total power of 100s of mW. We are unaware of a LED-array technology that achieves such high powers combined with micron-scale element size, ultra-fast switching, and large pixel counts.

- b. The images produced by Xu had only 1,024 pixels, while our system had 25,111 pixels. As mentioned in Xu's paper: "However, the dense LED array would cause a severe heating problem. In this experiment, the current was limited to 2A." It is therefore unclear how practical it would be to make a denser system with more pixels with currents in the 10s or 100s of amperes.
- c. The advantages of our scheme enabled us to produce meaningful microscope images of living organisms with high quality, whereas in the work of Xu, a very simple, large object was imaged. The difference in imaging capabilities is not a result of better engineering, but of the fundamental advantages of our approach, and the fact that LED-array technology (just like DMDs) is not mature enough to enable high-quality SPI.

We fully agree that Xu's work offers higher stability against vibrations, but this advantage has little practical meaning if the scheme itself does not offer all the benefits of SPI and is limited to the same imaging scenarios in which CCDs already operate extremely well. It is always preferable to have a monolithic imaging system, but there is no such option for rapid spatial modulation of arbitrary optical sources. Vibrations do represent a technical challenge, but they do not prohibit the use of our systems for microscopy, where a benchtop setup is used on an optical table. There are technical solutions to further reduce the effect of vibrations in more professional systems, e.g. dampeners, enclosures, or active motion cancelation. Furthermore, as discussed in the final paragraphs, the use of acousto-optic beam scanning as the next generation of our scheme could result in a device with no moving parts that is less susceptible to vibrations.

However, I think it meets the innovation criteria because the Cyclic Hadamard masks used here have by in-large been ignored by the research community and there is no experimental study showing their greatness. This is because mathematicians have been the ones considering the sampling patterns in terms of noise-suppression and undersampling efficiency, but they have not considered the experimental implementation of the masks. These Cyclic masks have the same noise suppression capabilities as ordinary Hadamard masks as well as Fourier masks (which is the best you can do with any sampling matrix due to their orthogonality as shown in ref 25 of the manuscript) and their reconstruction algorithms are of the same complexity but their undersampling efficiency is worse.

However, the experimental implementation of these cyclic masks is their greatest asset; and as this work shows one can now easily create a spatial 2D modulator with megahertz switch rates with nearly 100% modulation depth for many different types of waves. For example, for terahertz radiation they can use printed circuit board manufacturing techniques to create their spinning disk or for sound-waves they can just drill some holes. The fact that the number of image pixels is not easily changeable is also true of CCD imaging technology hence it is not a major disadvantage over other spatial light modulators, although it does need to be mentioned.

We fully agree with the reviewer that our experimental implementation represents a true innovation in the broader field of SPI, as it demonstrates how to exploit the mathematical concepts of cyclic masks to enable rapid SPI, and how the same concept can be applied to other wave technologies, e.g. ultrasound or terahertz. Indeed, we are currently working on an acoustic version of this concept using acoustically blocking masks to perform ultrasonic detection. Since there is no mature DMD technology for ultrasound, the use of cyclic Hadamard masks will enable new implementations of the SPI concept in ultrasound. Alternatively, one may use elements that convert from optics to the field of interest. In acoustic, these would be optical Fabry-Perots used to sense ultrasound, where the sensing region is the one that is illuminated. In terahertz imaging it would be elements such as highly resistive silicon wafers, in which the optical illumination produces spatial modulation of the terahertz beam.

Regarding the comment that our method is limited to a fixed number of pixels, it is only partially true. Our cyclic Hadamard patterns are produced on rings with a fixed diameter and a sub-millimeter width and we can easily produce tens of different patterns with different pixel counts on the same mask. In our implementation, we used 2 different geometries for the illumination pattern (square / circle), 2 different element geometries (square / hexagon) and 2 pixel counts options (25k / 10k), all in the same mask. Moreover, in contrast to DMDs, our patterns can have arbitrary pixel sizes. Indeed, changing between the patterns on mask would require realigning the beam, but this may be easily performed in a fully automated system. In terms of flexibility, the true limitation of using masks is not that the number (or size) of pixels is fixed in a given system, but rather that the different options need to be predetermined during production, i.e. it is not configurable. We now address those points in the main paper.

Until the authors have adequately discussed all the points in the above paragraph in the main manuscript this work will fail to show its greatness and discuss its place relative to other works. As such I can not recommend publication until this is done. Consequently, the manuscript title should be "A new spin on single-pixel imaging: spatial light modulation at megahertz switch-rates via cyclic Hadamard masks" because "cyclic codes" is a field related to cryptography hence creating confusion and "rapid" is ambiguous.

We have changed the title of the manuscript according to the reviewer's suggestions. We hope that we have convinced the reviewer that the importance of our work is not just in our ability to realize the innovative concept of cyclic codes in an experimental setup, but also in its practical utility for SPI as a generic method for rapid SPI.

Another issue is that they should say the improvement in signal-to-noise ratio (SNR) from their sub-pixel shifting procedure. For example, from the image in fig 2d they can obtain their SNR by taking the average of the signals transmitted through the large square and then divide it by the root-mean-

square of some completely black area. What is the difference in SNR if they only get an image from sub-pixel 3 and then from the average of all 5 sub-pixels (after their alignment procedure).

The SNR enhancement is now explained in detail in section 7 of the Supplementary Material. When the reconstruction error is solely due to noise at the output of the photodetector, the SNR should increase linearly with \sqrt{M} , where M is the number of frames that were averaged. However, our images included some minor structured artifacts due to inaccuracies in the mask motion, which are correlated between sub-pixel images and, thus, do not average out. For our fastest imaging rate example, shown in Fig. 3.b, we plotted the SNR gain as a function of the number of averaged sub-pixel images, M . An SNR gain of 2.6 was achieved for $M=33$, whereas the gain in the ideal case would be 5.74. We note that when imaging at higher speeds or lower powers, the effect of photodetector noise is expected to be more dominant with respect to structured, leading to higher SNR gains via averaging sub-pixel shifted images.

SNR gain of averaging M sub-pixel-shifted images in comparison to a single reconstruction for the image presented in Fig. 3b.

They also need to state their image pixel resolution and not just say $N = 25111$ in the figure captions, which I assume is something around a 50x50 pixel image but it is not exactly.

Our coding patterns were arranged for two types of illumination patterns, rectangle, and circle:

Circular pixels arrangement

Rectangle pixel arrangement

For the rectangle pattern, the size of the image is $P \times Q = N$ when $Q - P = 2$ based on the Quadratic Residue algorithm. **For the circular illumination, the pixels were arranged over the illuminated circle to maximize the number of pixels in the imaged area.** Therefore, some images in the paper are also presented in a circle. The 25,111 pixels would *approximately* correspond to a 158x158 image (not 50x50) if the pixels were arranged in a square grid. To avoid confusion, we mention for each image and video whether it was obtained with a hexagonal or square grid in sections 2 and 10 of the supplementary material and an explaining illustration was added to Sup Fig. 2.

They need to add scale bars to their videos and other images.

Scale bars have been added.

Supplementary figure 1, the tilt/tip mirror looks like it is just a normal mirror. They should add something to make it easily know this mirror is electrically controlled in sync with the rotation of the disc.

The figure has been changed accordingly.

REVIEWERS' COMMENTS

Reviewer #3 (Remarks to the Author):

In their reply, the authors have convinced me that this work is even more worthy of being published in Nature Communications. I recommend publication of the manuscript in its current form, although the authors may wish to add a few words to their abstract explicitly saying that their 72 FPS videos can be displayed in real-time.

Reviewer #3 comments:

In their reply, the authors have convinced me that this work is even more worthy of being published in Nature Communications. I recommend publication of the manuscript in its current form, although the authors may wish to add a few words to their abstract explicitly saying that their 72 FPS videos can be displayed in real-time.

Reply to reviewer #3

Thank you for recommending our work for publication and for the very useful feedback you gave us through the review process. We feel it have significantly improved the quality of our work. Following your advice, we have added some text addressing the capability of real time imaging.